# The Reproducibility and Usefulness of Estimated Average Glucose for Hyperglycemia Management during Health Checkups: A Retrospective Cross-Sectional Study

**DOI:** 10.3390/healthcare10050824

**Published:** 2022-04-29

**Authors:** Eun-Hee Nah, Seon Cho, Hyeran Park, Dongwon Noh, Eunjoo Kwon, Han-Ik Cho

**Affiliations:** 1Department of Laboratory Medicine and Health Promotion Research Institute, Korea Association of Health Promotion, Seoul 07572, Korea; dduddi3755@hanmail.net (S.C.); hyeran826@naver.com (H.P.); magu00@kahp.or.kr (D.N.); 4ever35@hanmail.net (E.K.); 2MEDIcheck LAB, Korea Association of Health Promotion, Seoul 07572, Korea; hanik@snu.ac.kr

**Keywords:** estimated average glucose, fasting plasma glucose, hemoglobin A1c, impaired fasting glucose, diabetes

## Abstract

HbA1c reflects average glucose levels over 3 months, but it does not measure glycemic variability. This study aimed to determine the reproducibility and usefulness of HbA1c-derived estimated average glucose (eAG) and to analyze the factors associated with eAG during health checkups. This cross-sectional retrospective study consecutively selected subjects who had undergone health checkups at 16 health-promotion centers in 13 Korean cities in 2020. The subjects comprised 182,848 healthy subjects with normoglycemia, 109,555 with impaired fasting glucose (IFG), and 35,632 with diabetes. eAG was calculated using Nathan’s regression equation. In all subjects, fasting plasma glucose (FPG) was found to be fairly strongly correlated with eAG (r = 0.811). When the subjects were divided into FPG subgroups, the strength of the correlation decreased among those with normoglycemia and IFG (*p* < 0.001). Higher eAG levels were associated with older age, females, higher FPG, and lower HDL-C and triglycerides (*p* < 0.05). The proportion of subjects with a higher value of FPG than eAG was 46.3% in poorly controlled diabetic patients, compared with only 1.5% in normoglycemic subjects. This suggests eAG could help patients to understand their glycemic variability intuitively and healthcare providers to identify patients who might worsen in hyperglycemia control through measuring the difference between eAG and FPG.

## 1. Introduction

Diabetes is a chronic illness caused by insulin resistance or poor insulin production. Adequate glycemic control within the optimal range is important for preventing diabetes-related microvascular or macrovascular complications [1]. Individuals with prediabetes constitute a high-risk group for diabetes, and so they also require awareness of their hyperglycemia and strict glycemic control to prevent progression into diabetes. Achieving this requires the participation of patients, including involvement in medical decision-making [2], mutual exchanges of information, and collaborating on lifestyle-change decisions. Fasting plasma glucose (FPG) and glycated hemoglobin (HbA1c) are the main indicators for monitoring chronic hyperglycemia in clinical settings.

HbA1c is considered the average glucose level over approximately 3 months. It is a simple and reproducible marker for long-term glycemic state assessments [3]. Its use has been recommended in the diagnosis for diabetes by the American Diabetes Association (ADA) [4]. Testing HbA1c is the primary method for assessing glycemic control and has a strong predictive value toward diabetes complications [5,6,7]. HbA1c is usually expressed as the percentage of glycated hemoglobin, whereas self-monitoring of blood glucose (SMBG), including FPG levels, is based on blood glucose levels expressed as milligrams per deciliter (mg/dL) or millimoles per liter (mmol/L). Estimated average glucose (eAG), expressed as milligrams per deciliter (mg/dL) or millimoles per liter (mmol/L), is much easier for patients to understand. In addition, it would be very practical if healthcare providers could predict mean glucose levels as an eAG level from a single blood sample test of HbA1c rather than through serial determination of glucose. Nathan et al. [8] conducted the International HbA1c-Derived Average Glucose (ADAG) Trial that included patients with type 1 diabetes, type 2 diabetes and without diabetes from ten international centers. Its results demonstrated the relationship between HbA1c and eAG and provided a method for calculating eAG using HbA1c levels. Nathan’s regression equation has been recommended for determining eAG by the ADA [9].

Although HbA1c was traditionally considered as the gold standard for assessing glycemic control, glycemic variability (GV) is a more meaningful measure of glycemic control than HbA1c in clinical practice [10]. GV, referring to oscillations in blood glucose, is usually defined by the measurement of fluctuations of glucose or other related parameters of glucose homeostasis over a given interval of time. Growing evidence demonstrated that GV was a significant and clinically meaningful glycemic metric and had drawn attention for its effects on adverse clinical outcomes, including diabetic macrovascular and microvascular complications, hypoglycemia and mortality [11,12,13]. Various metrics for the assessment of GV have been proposed. There is a lack of consensus about the appropriate method for characterizing it clinically despite extensive studies on glycemic variability [14,15]. Long-term GV is based on serial determinations over a longer period of time, involving HbA1c, serial FPG and postprandial glucose measurements. Measurements of long-term variability correlate with either mean concentration of blood glucose or mean HbA1c. Among those, the relationship between eAG and FPG has been investigated alongside diabetes in various studies [16,17,18]. However, those studies had limitations in their sample size, proportion of patients with prediabetes, age ranges, and hospital-based design. This study aimed to determine the reproducibility and usefulness of HbA1c-derived eAG and to analyze the factors associated with eAG in both diabetes and prediabetes among the general Korean population.

## 2. Materials and Methods

### 2.1. Study Subjects

This cross-sectional retrospective study consecutively selected subjects who had undergone health checkups at 16 health-promotion centers in 13 Korean cities between January 2020 and December 2020. The subjects comprised 182,848 healthy subjects with normoglycemia, 109,555 with impaired fasting glucose (IFG) and 35,632 with diabetes. The medical records of these subjects were also reviewed. Individuals with anemia (<13.0 g/dL and <12.0 g/dL in males and females, respectively), renal dysfunction (eGFR < 60 mL/min/1.73 m^2^) or pregnancy were excluded. The study protocol was reviewed and approved by the institutional review board (IRB) of the Korea Association of Health Promotion (approval no. 130750-202109-HR-007). This study was a retrospective study of medical records and all data were fully anonymized before authors accessed them and IRB waived the requirement for informed consent.

### 2.2. Laboratory Measurements

Venous blood was drawn during health checkups after an overnight fast, which included a complete blood count (CBC), biochemical measurements and HbA1c levels. The CBC and biochemical parameters were measured using the Sysmex XE-2100D analyzer (Sysmex, Kobe, Japan) and the Hitachi 7600 analyzer (Hitach, Tokyo, Japan), respectively. HbA1c levels were measured by ion-exchange high performance liquid chromatography using the Tosoh HLC-723 G8 analyzer (Tosoh Co., Tokyo, Japan), which are certified by the NGSP as being traceable to the Diabetes Control and Complications Trial. Serum FPG levels were determined by the hexokinase method [19]. eAG was calculated using Nathan’s regression equation: eAG (mg/dL) = 28.7 × HbA1c (NGSP, %) − 46.7 (eAG (mmol/L) = 1.59 × HbA1c (NGSP, %) − 2.59) [8].

### 2.3. Statistical Analyses

Statistical analyses were performed using SAS version 9.4 (SAS Institute, Cary, NC, USA). Pearson’s correlation analysis was used to determine the relationship between FPG and eAG. ANOVA and chi-square tests were used to compare eAG levels and the negative mean difference proportion between eAG and FPG according to FPG group. Multiple linear regression analysis was performed to determine the variables affecting increased eAG. Variables such as age, sex, waist circumference, hemoglobin, FPG, creatinine, HDL-C, LDL-C, triglycerides, eGFR and homeostatic model assessment-insulin resistance (HOMA-IR) were included in the multiple linear regression analysis. Scatter and distribution plots were developed to compare differences between eAG and FPG according to FPG group. A *p* value of <0.05 was considered statistically significant.

## 3. Results

### 3.1. Characteristics of Study Subjects and the Correlation between FPG and eAG According to FPG Group

Table 1 lists the characteristics of the 328,035 study subjects, including 182,848 with normoglycemia, 109,555 with IFG and 35,632 with diabetes. The mean age was 52 ± 13 years (range: 19–98 years). The study subjects were divided into four subgroups according to ADA criteria [4] and severity of diabetes: normoglycemia group, <100 mg/dL; prediabetes group, 100–125 mg/d; diabetes groups, 126–200 mg/dL, and >200 mg/dL. HbA1c levels were higher in the groups with higher FPG, especially in patients with diabetes with FPG > 200 mg/dL. The eAG level was 125.72 ± 15.95 mg/dL in the IFG group, while those in the diabetes groups with FPG 126–200 mg/dL and FPG > 200 mg/dL were 161.07 ± 30.03 mg/dL and 254.6 ± 52.97 mg/dL, respectively. Overall, FPG had a fairly strong correlation with eAG (r = 0.811, *p* < 0.001) (Figure 1). However, when subjects were divided into subgroups, the correlation coefficients decreased: normal FPG, r = 0.210; IFG group, r = 0.413; FPG = 126–200 mg/dL, r = 0.573; and FPG > 200 mg/dL, r = 0.524.

### 3.2. Mean Glucose Levels for Specified HbA1c Levels

Table 2 lists the mean FPG and eAG values for specific HbA1c levels based on the present study and ADAG data. Mean FPG levels for specified HbA1c levels obtained in our study tended to be lower than those obtained by ADAG data. 

### 3.3. Difference between eAG and FPG Values

The difference between eAG and FPG values decreased in groups with higher FPG, especially among the patients with poorly controlled diabetes (FPG > 200 mg/dL) (Table 1). Furthermore, 46.3% of subjects had higher FPG than eAG, resulting in a negative mean difference between eAG and FPG among patients with poorly controlled diabetes, compared with only 1.5% of healthy subjects with normoglycemia (Table 3) (Figure 2).

### 3.4. Factors Associated with eAG

In the multiple regression model, older age, female sex, and low high-density lipoprotein cholesterol (HDL-C) and triglycerides were associated with eAG (all *p* < 0.05). However, HOMA-IR was not associated with eAG (Table 4).

## 4. Discussion

This study indicated that FPG is strongly correlated with eAG; however, this correlation weakened among patients with normoglycemia and IFG. Mean FPG and eAG values for specified HbA1c levels obtained from our population tended to be lower than those obtained from ADAG data. We further demonstrated that eAG was associated with older age, female sex, and low HDL-C and triglycerides. In uncontrolled hyperglycemia, the difference between eAG and FPG decreased or had a negative mean difference. This suggests it may help healthcare providers to identify patients who may be in poorly controlled hyperglycemia by measuring the difference between eAG and FPG.

HbA1c reflects average glucose levels over 3 months, but it does not measure glycemic variability. For patients with diabetes, glycemic control is best evaluated using a combination of SMBG and HbA1c values [21]. Long-term GV is based on serial determinations over a longer period of time, involving HbA1c, serial FPG and postprandial glucose measurements. On the other hand, those who receive health checkups at health-promotion centers do not have SMBG data. They pay more attention to their FPG and HbA1c values. We used Nathan’s regression equation to calculate the eAG levels of our study subjects and investigated their association with corresponding FPG levels. This indicated a fairly strong association between eAG and FPG levels, which decreased in individuals with normoglycemia and prediabetes. This may be due to the higher contribution of postprandial glucose in individuals with normoglycemia and prediabetes as both FPG and postprandial glucose levels determine the eAG level. The finding that the correlation between eAG and FPG weakened in well-controlled diabetes was consistent with previous studies [16,18].

The differences between eAG and FPG were also investigated among groups with different glucose levels. The differences between eAG and FPG decreased in cases of severely uncontrolled diabetes. This suggests that FPG contributed to eAG more than postprandial and bedtime glucose values among patients with severely uncontrolled diabetes. This finding also reflected a strong correlation between eAG and FPG in this group. Furthermore, some subjects had a negative mean difference between eAG and FPG. Among the subjects with poorly controlled diabetes, 46.3% had higher FPG than eAG, compared with only 1.5% of subjects with normoglycemia. High morning blood glucose in patients with diabetes can be explained by the dawn phenomenon. The dawn phenomenon refers to an increase in blood glucose levels or a rise in the amount of insulin needed to maintain normoglycemia that occurs in the absence of antecedent hypoglycemia or waning insulin levels in the early morning. The pathogenic mechanism is nocturnal surges in growth hormone secretion and insufficient repression of insulin-antagonistic hormone secretion due to the decrease in insulin. Another considerably rarer cause of morning hyperglycemia is known as the Somogyi effect. This is the body’s response to low blood glucose at night. The body produces more glucose to compensate, leading to high blood glucose in the early morning [22,23,24]. Endogenous or exogenous insulin is less effective in poorly controlled hyperglycemia than in well-controlled patients, which might lead to severe fasting hyperglycemia, resulting in higher FPG than eAG levels. This finding suggests that when the difference between eAG and FPG decreased or had a negative mean difference, hyperglycemia control worsened in not only severe diabetes but also in some of the prediabetes cases.

Frequent discordances have been reported between eAG and SMBG levels [20,25,26]. There have been some discrepancies in eAG levels across study populations, diabetes types, glucose monitoring methods, ages, and races. We estimated the mean FPG for specified HbA1c levels among our subjects. The mean FPG and mean eAG levels we obtained tended to be lower than those obtained from reanalyzed ADAG data. Wei et al. [27] calculated average glucose concentrations using only the blood glucose values collected by SMBG for specified ranges of HbA1c values obtained from subjects in the ADAG study. Only SMBG values were used where they could set a meal-related time (i.e., fasting, preprandial, postprandial, and bedtime) instead of combining all the 7-point SMBGs from the ADAG group and continuous glucose monitoring data into a single data point to obtain a single eAG value. The difference in mean FPG between our data and reanalyzed ADAG data may be attributable to differences in study populations, disease severity, glucose measurement method, or sample sizes.

We determined the factors associated with eAG among our subjects. Female sex was a factor associated with eAG in our study. eAG values were higher in females with prediabetes and diabetes. The effect of sex on eAG is controversial. One study indicated that eAG values were lower in females [16], while another [17] did not find any significant difference between the eAG values of the two sexes. In addition to sex, age, HDL-C, and triglycerides were also associated with eAG. The eAG was derived from HbA1c in this study. HbA1c levels are not only associated with diabetes, but also with a variety of other factors such as age, body weight, and blood lipids [28]. Furthermore, a study [29] reported that higher average glucose levels may be a risk factor for stroke, even among those without a diabetes diagnosis.

Our study has some limitations. First, we could not obtain postprandial and bedtime glucose values, since glucose values were obtained from health checkup data. Venous blood was drawn during health checkups after an overnight fast. Second, information on diabetes medication or insulin therapy and type of diabetes could not be obtained. Nevertheless, the present study included large numbers of both diabetics and prediabetics nationwide and is also based on a community-based health checkup cohort which reflects the real-world data.

## 5. Conclusions

Though the relationship between eAG and FPG has been previously presented in other studies, this study tried to demonstrate reproducibility and usefulness of the HbA1c-derived eAG in the general population. In terms of public healthcare, use of HbA1c-derived eAG is an affordable method to improve self-management in individuals with hyperglycemia. FPG along with HbA1c-derived eAG could be helpful for both diabetic and prediabetic patients without SMBG data in a more intuitive way. In addition, we suggest that the difference between eAG and FPG may depend on glycemic control in individuals with hyperglycemia. Healthcare providers could also give information to patients who may be in poorly controlled hyperglycemia by measuring the difference between eAG and FPG in an easy way.

## Figures and Tables

**Figure 1 healthcare-10-00824-f001:**
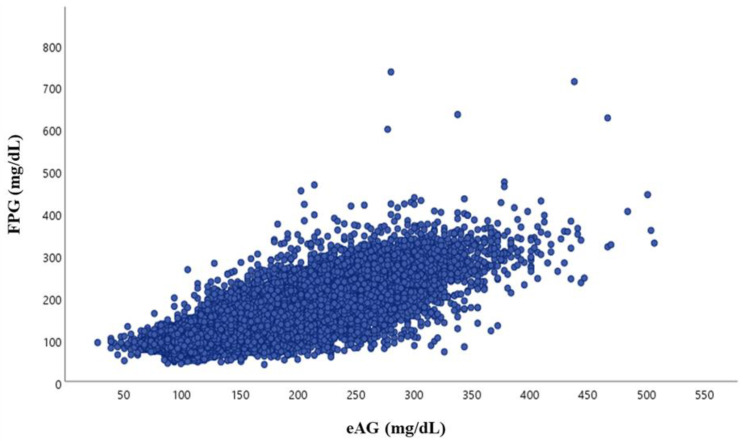
Relationship between estimated average glucose (eAG) and fasting plasma glucose (FPG) in the study subjects.

**Figure 2 healthcare-10-00824-f002:**
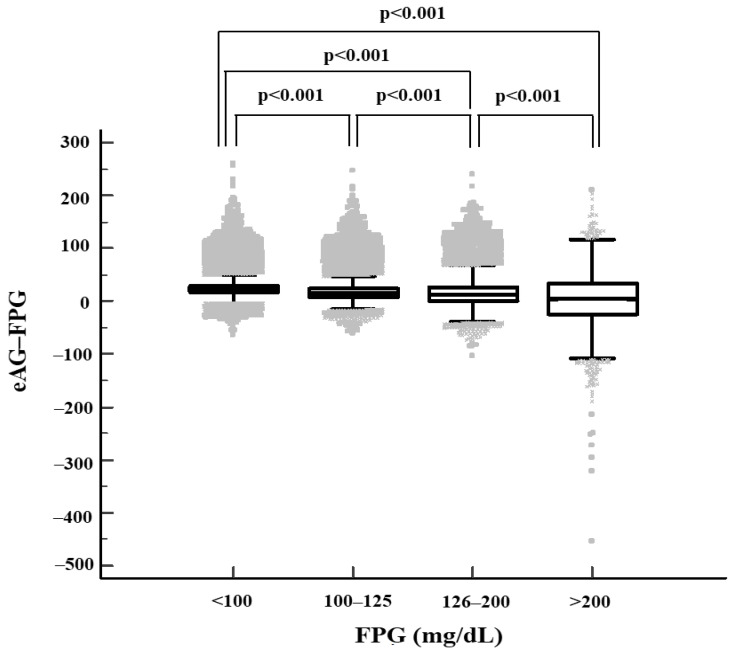
Scatter and distribution plots of differences between eAG and FPG in the study subjects. Box limits and horizontal lines within boxes represent interquartile ranges and the median, respectively. The upper and lower whiskers indicate the 97.5th and 2.5th percentiles, respectively. *p* value is from Wilcoxon rank sum test. eAG–FPG: Difference between eAG and FPG.

**Table 1 healthcare-10-00824-t001:** Characteristics of study subjects and correlations between fasting plasma glucose (FPG) and estimated average glucose (eAG) according to FPG group.

	All		FPG								*p* Value	Post-Hoc
			<100 mg/dL ^a^		100–125 mg/dL ^b^		126–200 mg/dL ^c^		>200 mg/dL ^d^			
Total												
Subj., N	328,035		182,848		109,555		31,875		3757			
Age, year	52±	13	49±	13	56±	12	58±	11	53±	12	<0.001	a < d < b < c
Hb, g/dL	14.3±	1.5	14.1±	1.6	14.5±	1.5	14.7±	1.5	15.3±	1.6	<0.001	a < b < c < d
FPG, mg/dL	104±	25	90±	6	109±	7	145±	18	250±	44	<0.001	a < b < c < d
HbA1c, %	5.9±	0.9	5.6±	0.4	6.0±	0.6	7.2±	1	10.5±	1.8	<0.001	a < b < c < d
eAG, mg/dL	123.81±	25.74	113.49±	10.91	125.72±	15.95	161.07±	30.03	254.60±	52.97	<0.001	a < b < c < d
Corr.coeff.(r)	0.811		0.21		0.413		0.573		0.524			
eAG–FPG	20.24±	15.57	23.31±	11.44	16.97±	14.53	15.78±	24.62	4.14±	47.96	<0.001	d < c < b < a
Male												
Subj., N	172,095		83,791		64,422		21,272		2610			
Age, year	52±	13	48±	13	54±	12	57±	10	52±	11	<0.001	a < d < b < c
Hb, g/dL	15.3±	1.2	15.3±	1.1	15.3±	1.2	15.3±	1.3	15.8±	1.3	<0.001	a,b,c < d
FPG, mg/dL	107±	27	91±	6	109±	7	146±	18	250±	45	<0.001	a < b < c < d
HbA1c, %	6.0±	1	5.6±	0.4	6.0±	0.6	7.2±	1	10.4±	1.8	<0.001	a < b < c < d
eAG, mg/dL	125.68±	27.76	113.65±	10.97	124.78±	15.99	160.24±	29.99	252.52±	53.09	<0.001	a < b < c < d
eAG–FPG	18.80±	16.6	22.75±	11.8	15.75±	14.57	14.43±	24.7	2.73±	48.56	<0.001	d < c < b < a
Corr.coeff.(r)	0.817		0.142		0.413		0.568		0.52			
Female												
Subj., N	155,940		99,057		45,133		10,603		1147			
Age, year	53±	13	49±	13	58±	11	60±	10	54±	12	<0.001	a < d < b < c
Hb, g/dL	13.2±	1.1	13.1±	1.1	13.4±	1.1	13.5±	1.2	14.0±	1.4	<0.001	a < b < c < d
FPG, mg/dL	100±	22	90±	7	108±	7	144±	17	252±	42	<0.001	a < b < c < d
HbA1c, %	5.9±	0.8	5.6±	0.4	6.1±	0.6	7.3±	1	10.7±	1.8	<0.001	a < b < c < d
eAG, mg/dL	121.75±	23.13	113.35±	10.85	127.05±	15.8	162.73±	30.04	259.34±	52.43	<0.001	a < b < c < d
Corr.coeff.(r)	0.801		0.264		0.425		0.591		0.535			
eAG–FPG	21.84±	14.17	23.78±	11.1	18.72±	14.3	18.48±	24.24	7.35±	46.43	<0.001	d < b,c < a

Data are mean ± standard deviation values, except where indicated otherwise. *p* values are from ANOVA with Scheffe post-hoc analysis. Correlation coefficients are for correlations between eAG and FPG. eAG–FPG is difference between eAG and FPG. Abbreviations: Hb, hemoglobin; HbA1c, glycated hemoglobin; Corr.coeff., Correlation coefficient; Subj., subject number.

**Table 2 healthcare-10-00824-t002:** Mean glucose levels for specified HbA1c levels from the present study and A1c-derived average glucose (ADAG) data.

HbA1C	Estimated Mean Glucose Concentration											Estimated Mean FPG										
	ADAG Study *						Present Study					Reanalyzed ADAG Data **						Present Study				
(%)	mg/dL			mmol/L			mg/dL			mmol/L		mg/dL			mmol/L			mg/dL			mmol/L	
≤5.6							107.4±	6.2		5.9±	0.3							92.6±	9.4		5.1±	0.5
6.0	126	(100–152)		7.0	(5.5–8.5)																	
5.7–<6.5							124.1±	6.0		6.9±	0.3	122	(117–127)		6.8	(6.5–7.0)		101.4±	12.7		5.6±	0.7
6.5–<7.0							144.7±	4.0		8.0±	0.2	142	(135–150)		7.9	(7.5–8.3)		120.8±	17.3		6.7±	1.0
7.0	154	(123–185)		8.6	(6.8–10.3)																	
7.0–<7.5							159.2±	4.0		8.8±	0.2	152	(143–162)		8.4	(7.9–9.0)		132.8±	21.2		7.4±	1.2
7.5–<8.0							173.6±	4.0		9.6±	0.2	167	(157–177)		9.3	(8.7–9.8)		143.8±	26.1		8.0±	1.4
8.0	183	(147–217)		10.2	(8.1–12.1)																	
8.0–8.5							189.3±	4.9		10.5±	0.3	178	(164–192)		9.9	(9.1–10.7)		156.1±	32.7		8.7±	1.8
9.0	212	(170–249)		11.8	(9.4–13.9)																	
10.0	240	(193–282)		13.4	(10.7–15.7)																	
11.0	269	(217–314)		14.9	(12.0–17.5)																	
12.0	298	(240–347)		16.5	(13.3–19.3)																	

Data are mean ± standard deviation or (95% CI) values. * Data were obtained from ADAG study [8]. ** Data were obtained from reanalyzed ADAG data [20].

**Table 3 healthcare-10-00824-t003:** Differences between eAG and FPG in the study subjects.

		FPG								*p* Value
		<100 mg/dL		100–125 mg/dL		126–200 mg/dL		>200 mg/dL		
eAG–FPG *	Negative, N (%)	2681	(1.5)	10,582	(9.7)	8234	(25.8)	1740	(46.3)	<0.001
	Positive, N (%)	180,167	(98.5)	98,973	(90.3)	23,641	(74.2)	2017	(53.7)	

*p* value is from chi-square test. * eAG–FPG: Difference between eAG and FPG.

**Table 4 healthcare-10-00824-t004:** Factors associated with eAG in the study subjects.

	Coefficient	(SE)	Standard Coefficient	*p* Value
Age	0.256	(0.036)	0.119	<0.001
Sex (ref. male)	4.168	(1.401)	0.078	0.003
WC	−0.017	(0.048)	−0.006	0.727
Hb	0.003	(0.352)	0.000	0.993
FPG	0.906	(0.018)	0.817	<0.001
Creatinine	1.292	(3.228)	0.011	0.689
HDL-C	−0.072	(0.033)	−0.036	0.031
LDL-C	0.015	(0.011)	0.020	0.170
TG	−0.010	(0.003)	−0.046	0.003
e-GFR	0.055	(0.044)	0.030	0.213
HOMA-IR	0.173	(0.224)	0.012	0.440

Durbin–Watson D = 1.767. Adjusted R^2^ = 0.721. WC, waist circumference; HDL-C, high-density lipoprotein cholesterol; LDL-C, low-density lipoprotein cholesterol; TG, triglycerides; e-GFR: estimated Glomerular filtration rate; HOMA-IR, homeostasis model assessment of insulin resistance; SE, standard error.

## Data Availability

Not applicable.

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
