# Peer review of "The Reproducibility and Usefulness of Estimated Average Glucose for Hyperglycemia Management during Health Checkups: A Retrospective Cross-Sectional Study"

_healthcare, 2022, doi:10.3390/healthcare10050824_

Round 1

Reviewer 1 Report

In this study, Nah et al studied the reproducibility of the HbA1C-derived estimated average glucose (eAG) method of assessing diabetes. For this, they performed a retrospective cross-sectional study where they selected subjects with normoglycemia, impaired fasting glucose levels, and diabetes. They observed that the eAG for strongly associated with fasting plasma glucose (FPG) levels. Also, higher eAG was specifically associated strongly with older females with higher FPG and lower triglycerides. Overall, they conclude that eAG could help patients to understand their glycemic variability. Although this study has a strong sample size, the authors need to address the following concerns:

  1. The major limitation of this study is the lack of novelty and a strong rationale. These are established facts with regard to the correlation between FPG and eAG. Furthermore, checking the “reproducibility” of a method or a finding is considered to be supplemental in most of the studies. So, unless the authors provide some novel analyses of the data and address the rationale, this study is difficult to be acceptable.

  1. HbA1C is a globally accepted method of assessing glycemic control. The authors posit that the “estimated average glucose (eAG) is more practical and much easier for patients to understand (Lines 48-49)”. The authors need to explain in detail why do they think it will be a better method? Secondly, since it is derived from HbA1C, why would the patients/healthcare providers want to do the calculations when they can refer to the accepted range?

  1. The authors talk frequently about glycemic variability. They need to define and explain what do they exactly mean by glycemic variability, the importance of this parameter in understanding the disease progression, and how exactly does eAG help to do that.

  1. Lines (108-110)- The authors mention “However, when subjects were divided into subgroups, the correlation coefficients decreased: normal FPG, r=0.210; IFG group, r=0.413; FPG=126–200 mg/dL, r=0.573; and FPG >200 mg/dL, r=0.524.”- the author need to explain this sub-grouping and discuss why do they think they see a reduced correlation.

  1. The authors need to do an extensive revision of the article to make it reader-friendly. There is a random inclusion of concepts, terminologies, and phenomena that can be confusing.

  1. Finally, the authors conclude by saying “It may also help healthcare providers for better identifying patients who might worsen in hyperglycemia control by measuring the difference between eAG and FPG”. Although this might be true, the authors should explain how exactly will the difference between eAG and FPG inform about future glycemic outcomes. Considering the fact that hyperglycemia can be caused by multiple factors, how would eAG and FPG calculations provide more insight than HbA1C aloe?

Minor:

  1. The authors frequently mention that “The subjects comprised 182,848 patients with normoglycemia, 109,555 with impaired fasting glucose (IFG) and 35,632 with diabetes”. In this “patients with normoglycemia” can be confusing. The authors need to clarify whether these are true “patients” with some disease or whether they are healthy subjects with normal glycemic levels. Also, although the sample size is impressive, they do not need to repeat the statement multiple times.

  1. Line 75-76: The authors need to clarify what do they mean by “the analysis used anonymous clinical data.”

  1. Line 84-85: “Serum FPG 84 levels were determined using the hexokinase method”. Citation needed.

  1. The authors need to mention which posthoc analysis was performed for ANOVA.

  1. Tables 1 and 2 need to be reformatted for easier reading.

Reviewer 2 Report

In this cross-sectional retrospective study, Nah et al. investigated the reproducibility and usefulness of HbA1c-derived estimated average glucose (eAG) and analyzed the factors associated with eAG in both diabetes and prediabetes among 182,848 patients with normoglycemia, 109,555 with impaired fasting glucose (IFG), and 35,632 with 16 diabetes.

Authors report that fasting plasma glucose is strongly correlated with eAG, and this correation weakened among patients with normoglycemia and impaired fasting glucose. Authors further demonstrated that eAG was associated with older age, female sex, and low HDL-C and triglycerides.

Interesting work, only minor issues need to be addressed.

MINOR ISSUES:

  1. Minor English editing.
  2. Introduction: lines 46-47, please clarify this sentence
  1. Discussion:

-lines 201-205, please add a possible explanation on the other factors associated with the eAG observed in the study.

-lines 206-216, please add in the limitation that the reproducibility and usefulness of eAG cannot be extend to individuals with anemia, renal dysfunction and pregnancy.

-could glucose lowering drugs impact on the observed results?

Round 2

Reviewer 1 Report

The authors have addressed most of the comments. For final submission, the authors need to do thorough proofreading of the manuscript to avoid minor grammatical errors (spelling and punctuation).